# Serotonin and Dopamine Blood Levels in ADHD-Like Dogs

**DOI:** 10.3390/ani13061037

**Published:** 2023-03-13

**Authors:** Ángela González-Martínez, Susana Muñiz de Miguel, Noemi Graña, Xiana Costas, Francisco Javier Diéguez

**Affiliations:** 1VTH Rof Codina, Santiago de Compostela University, 27002 Lugo, Spain; 2Lar do Belelle, Canine Center, 15509 Fene, Spain; 3Etoloxía, Behavior Medicine Service, 36312 Pontevedra, Spain; 4Anatomy, Animal Production and Clinical Veterinary Sciences Department, Santiago de Compostela University, 27002 Lugo, Spain

**Keywords:** attention deficit, canine, hyperactivity behavior, Lasso regression, neurotransmitters

## Abstract

**Simple Summary:**

Attention deficit hyperactivity disorder (ADHD) is a relatively common neurodevelopmental disorder mainly affecting children and adolescents. ADHD is associated with significant social stigma, psychosocial adversities, and reduced working abilities. As with humans, dogs can suffer from ADHD-like behaviors, but to date, there are few studies on this condition in the canine species. Therefore, the present study analyzed the levels of serum serotonin and dopamine in dogs with signs that are similar to the ones presented in human patients with ADHD (compared to control dogs) that were assessed clinically and through different behavior scales. The results obtained indicate that both neurotransmitters have a tendency to be observed in lower concentrations in dogs presenting behavioral signs consistent with ADHD-like disorders. This finding may be useful for contributing to the study of the diagnosis and treatment of this disorder in dogs. ADHD-like associated behaviors can disrupt the normal coexistence of the dogs in their homes and contribute to abandonment, which continues to be a major social and animal welfare problem today.

**Abstract:**

As with humans, dogs can suffer from attention deficit hyperactivity disorder-like (ADHD-like) behaviors naturally and exhibit high levels of hyperactivity/impulsivity and attention deficit problems, making the domestic dog a potential animal model for ADHD. ADHD has a very complex pathophysiology in which many neurotransmitters are involved, such as serotonin and dopamine. The aim of the study was to evaluate serum serotonin and dopamine levels in dogs with ADHD-like symptomatology. Fifty-eight dogs were studied, of which, thirty-six were classified as ADHD-like after physical and behavioral assessments. Additionally, the dogs’ owners performed a series of scientifically validated questionnaires which included C-BARQ, the Dog Impulsivity Assessment Scale, and the Dog-ADHD rating scale. Serum from every animal was collected after the behavioral assessments and analyzed with commercial ELISA tests for serotonin and dopamine determination. Kruskal–Wallis tests and Lasso regressions were applied to assess the relationships between both neurotransmitters and the ADHD-like behaviors (as assessed by clinical evaluation and through the different questionnaires). The dogs clinically classified as ADHD-like showed lower serotonin and dopamine concentrations. Further, serotonin and dopamine levels were also linked to aggression, hyperactivity, and impulsivity. Decreased serotonin concentrations were also related to fear, attachment, and touch sensitivity. Finally, it must be noted that our data suggested a strong relationship between serotonin and dopamine and ADHD-like behaviors.

## 1. Introduction

ADHD (attention deficit hyperactivity disorder) is one of the most frequently diagnosed syndromes in child psychiatry. The key symptoms (inattentiveness, impulsivity, and hyperactivity) deteriorate the relationships of these children both in their families and with other children, thus increasing the risk of social isolation [1]. The diagnostic and statistical manual of mental disorders (DSM-5) has established that these signs should be present for at least 6 months [2]. Furthermore, two-thirds of patients with ADHD diagnoses in childhood continue to have persistent, impairing symptoms in adulthood [3,4].

Neuroimaging studies have shown structural alterations in several brain regions in children and adults with the condition. The volumes of the accumbens, amygdala, caudate, hippocampus, putamen, and intracranial region were smaller in individuals with ADHD compared with the controls because of a delay in maturation. These data confirm that patients with ADHD have altered brains and, therefore, ADHD is a brain disorder and a relevant model of a disorder of brain maturation delay [4,5,6,7].

As with humans, dogs can suffer from ADHD-like behaviors naturally and manifest high levels of hyperactivity/impulsivity and attention deficit problems, making the domestic dog a highly potential animal model for ADHD [8,9,10].

ADHD is a very complex pathophysiologic disorder in which many neurotransmitters are involved. GABA, glutamate, histamine, serotonin, and dopamine are involved in different ways in ADHD. ADHD has been associated with dopamine dysfunction in the mesocortical, mesolimbic [11], and nigrostriatal pathways [12]. Disturbances in the mesocortical dopamine pathway are thought to give rise to cognitive deficits. The mesocortical pathway has dopamine cell bodies in the ventral tegmental area (VTA) with projections via the medial forebrain bundle to the frontal cortex. As such, it acts as an important regulator of cognitive functioning. Disturbances in the mesolimbic pathway are thought to cause motivational deficits in ADHD. The mesolimbic dopamine pathway forms a crucial part of the brain “reward” circuits. Here, fibers from the VTA project via the medial forebrain bundle to discrete parts of the limbic system, especially to the nucleus accumbens [11]. Finally, the dopaminergic nigrostriatal pathway has projections from the substantia nigra to the striatum and plays a critical role in dopamine signaling [13], particularly, that involved in cognitive and voluntary movement control [14].

Methylphenidate, a stimulating drug, inhibits the presynaptic uptake of dopamine. Approximately 70% of children and adolescents with ADHD respond positively to stimulating treatment [1,15], and it seems that some ADHD-like dogs also respond to this therapy [16,17]. Some genetic markers for the dopamine system are found in humans [1,18] and dogs. Thereby, polymorphism in the tyrosine hydroxylase gene is associated with activity-impulsivity in German Shepherd dogs [19]. Likewise, DRD4 exon 3 polymorphism is associated with significantly higher scores in the activity-impulsivity dimension of a dog-ADHD rating scale in police German Shepherds [20]. Thus, dopamine dysfunction has been strongly related to ADHD pathophysiology.

Nevertheless, approximately 30% of ADHD humans cases are ‘non-responders’ to the first line of treatment, i.e., methylphenidate or amphetamine [21]. Studies of animal models of ADHD indicate an intimate interplay between serotonin and dopaminergic neurotransmission. Serotonin behavioral effects are complex and unspecific. Serotonin plays a role in the regulation of mood, eating control, sleep, arousal, and pain regulation [22]. Markedly, serotonin regulates dopamine transmission via multiple serotonin receptors [23], and therefore, it is plausible that a serotonin disturbance could also contribute to ADHD-associated behavioral and cognitive impairments in part via its action on dopamine. Different gene polymorphisms that modulate the expression of serotonin transporters and serotonin receptors which, in turn, modulate extracellular serotonin levels have been found to increase the susceptibility to ADHD [14]. Drugs that inhibit the reuptake of serotonin are prescribed in the treatment of different mental illness such as depression, anxiety disorders, and obsessive-compulsive disorder [22]. Selective serotonin re-uptake inhibitors and non-stimulant drugs acting on the serotonin system are, however, clinically effective for treating ADHD [24]. Further, in dogs, fluoxetine was useful and well tolerated in treating ADHD-like behaviors [25,26,27].

In neurocognitive models of ADHD, ‘inhibition’ is considered to be a core deficit that is mediated by serotonin [24,28]. Previous studies have indicated a decrease in blood serotonin levels among hyperactive and ADHD children [24,29,30]. In fact, a chronic deficiency in available serotonin may contribute to the clinical symptoms of ADHD [31]. Interestingly, lower levels of urine serotonin and dopamine were found in dogs with higher impulsivity [32]. In addition, aggressive and impulsive dogs showed lower levels of blood serotonin [33,34,35,36]. Nevertheless, there are no previous studies assessing serotonin or dopamine levels in ADHD-like dogs.

Therefore, the aim of the present study was to evaluate serum serotonin and dopamine levels in ADHD-like dogs (vs control dogs) as assessed by clinical evaluation and in dogs assessed through different validated questionnaires.

## 2. Materials and Methods

### 2.1. Surveyed Animals and Data Collection

The study was carried out at the Rof Codina Veterinary Teaching Hospital (VTHRC), which is part of the Veterinary Faculty of Lugo (the referral center for veterinary clinics in northwest Spain), with the collaboration of two other leading behavioral medicine services, Lar do Belelle and Etoloxía, also in northwest Spain. A total of 58 dogs were included in the study. They were recruited through cases that came to behavioral medicine services because of hyperactivity, impulsivity, or attention deficit problems and through an active search of dogs with ADHD-like signs, and the control dogs were recruited through social media networks such as Facebook, Instagram, and LinkedIn. We also shared an e-mail chain, starting with our personal contacts. Therefore, 40 of the 58 dogs had attended consultation at the VTH RC, with 10 at the Lar do Belelle center and 8 at the Etoloxía center. During consultation, we collected data on gender, entire or neutered, age (months), age at acquisition (months), weight (kg), activity pattern (whether the dog performed some activity such as agility, dog jogging, dog trekking, dog biking, mushing, or other), whether the owner had had other dogs previously, and habitat (rural or urban). Evaluations of the dogs were carried out by a veterinarian specializing in behavioral medicine.

### 2.2. Dog Assessment

As mentioned, all the dogs were examined by a veterinarian specializing in behavioral medicine. A complete history and behavioral and physical examinations (with the collection of blood samples) was carried out. ADHD-like were suspected in cases that presented hyperactivity, impulsivity, or attention deficit symptoms (a list of different symptoms of the ADHD-like dogs are described in Table 1) [8,25,37].

Based on the above and on the Canine Behavioral Assessment and Research Questionnaire (C-BARQ) [38,39,40], 5 dogs with suspected attention deficit related to phobias, hyperactivity that could be associated with a lack of exercise, and impulsivity that could originate from previous learnings, as well as patients with organic pathologies (including chronic pain), were not included in the final sample of 58 animals.

In addition, all the dogs’ owners completed a series of scientifically validated questionnaires in addition to the C-BARQ: Dog Impulsivity Assessment Scale (DIAS) [41,42] and the Dog-ADHD rating scale, owner’s version [8].

The C-BARQ questionnaire consists of 100 questions describing the different ways in which dogs typically respond to common events, situations, and stimuli in their environments. Responses are grouped into 14 behavioral traits: stranger-directed aggression, owner-directed aggression, dog-directed aggression, dog-directed fear, familiar dog aggression, trainability, chasing, stranger-directed fear, nonsocial fear, separation-related problems, touch sensitivity, attachment/attention seeking, excitability, and energy. All traits are expressed on a 0 to 4 scale in which 0 indicates no sign of such behavior and 4 indicates a severe form of the behavior.

The DIAS is an owner-based questionnaire with 18 statements concerning their perceptions of the behavioral tendencies of their dog, with individual item responses scored on a 1–5 scale. The scores were calculated for the overall questionnaire (OQS) and for each of the three subscales that compose it: “Behavioural Regulation”, “Aggression & Response to Novelty”, and “Responsiveness” [32,41,42,43].

The Dog-ADHD rating scale, owner version (Dog ARS) was developed on the basis of a validated human ADHD questionnaire. In the Dog ARS, six items are designed to refer to attention deficits in a dog (Subscale I) and seven are designed to measure the level of motor activity and impulsivity (Subscale II). The owners have to choose answers from the presented alternatives representing different frequencies of how often the statement is true for their dog on a four-level scale (never, rarely, often, or very often) [8].

The dogs were treated according to the European and Spanish legislation on animal protection (Directive 86/609/EEC, Real Decreto 1201/2005), and the Ethical Committee of the VTH Rof Codina approved the experiments and procedures.

### 2.3. Serotonin and Dopamine Measurement

Serum from every animal was collected after the behavioral assessment. Blood samples were obtained by venipuncture using tubes without anticoagulant from the jugular, cephalic, or saphenous vein. After that, they were centrifuged and frozen at −80 °C until analysis.

The serum samples were analyzed with two commercial ELISA tests (the Serotonin High Sensitive ELISA, DLD Diagnostika GMBH, Hamburg, Germany), and the Canine Dopamine DA ELISA Kit, MyBioSource, San Diego, CA, USA). These are competitive ELISA kits for the quantitative determination of serotonin and dopamine, respectively.

The analyses were performed following the recommendations and procedures indicated by the manufacturer for serum samples. According to the test, concentrations were expressed in ng/mL.

### 2.4. Statistical Analyses

Statistical analyses were performed using R Statistical Software (Appendix A). Initially, the scores obtained for the studied animals in the different traits for the DIAS, Dog ARS, and C-BARQ questionnaires were transformed into binary variables, taking as cut-off points for each trait the median of the study population for that trait. This allowed for dividing the population into two groups of similar sizes. Subsequently, the differences in the serotonin and dopamine concentrations for the suspected ADHD dogs, based on the behavioral examinations (yes/no) and the different DIAS, Dog ARS, and C-BARQ tests (scores under or over the median), were examined by Kruskal–Wallis tests.

Finally, a Lasso regression (glmnet package, https://cran.r-project.org/web/packages/glmnet/index.html, accessed on 28 June 2021) was applied to evaluate the effects of serotonin and dopamine levels on the suspected diagnosed ADHD dogs (yes/no) and the different DIAS, Dog ARS, and C-BARQ tests (as mentioned, and transformed into the binary variables of scores above or over the median). Gender, reproductive status (entire or neutered), age (months), age at acquisition (≤2 months or >2 months of age), weight (kg), activity pattern (whether the dog performed some activity such as agility, dog jogging, dog trekking, dog biking, mushing, or other), whether the owner had had other dogs previously, and habitat (rural or urban) were also included in the models as independent control variables. One model was performed for the ADHD-like suspected symptoms and one for each DIAS, Dog ARS, and C-BARQ test.

A K-fold cross-validation was performed to determine the lambda value that produced the lowest test mean squared error. The main advantage for the Lasso regression was that the model performed automatic feature selection to decide which variables should and should not have been included on their own. It helps to reduce overfitting and it is useful for variable selection when there are several independent variables that may not contribute to the outcome.

## 3. Results

After the behavioral examinations, 36 of the 58 dogs (62.0%) were classified as ADHD-like (presenting hyperactivity, impulsivity, or attention deficit symptoms). The remaining ones were used as control dogs (no ADHD-like symptoms). Descriptive features of the studied population (total and stratified by ADHD-like diagnosis) are shown in Table 2.

The mean serotonin and dopamine serum concentrations in the dogs with suspected diagnoses as being ADHD-like (vs the not suspected ADHD-like dogs) and the dogs with scores over and under the median for the different DIAS, Dog ARS, and C-BARQ traits evaluated are provided in Table 3. In the univariate approach (Kruskall–Wallis test), the differences between the groups for the serotonin serum concentrations were significant for the suspected ADHD-like dogs, the DIAS total score, the DIAS behavioral regulation score, the DIAS aggression score, the Dog ARS activity-impulsivity score, and the C-BARC owner-directed aggression, dog-directed fear, and non-social fear scores. In the case of dopamine, significant differences were observed for the dogs with a suspected diagnosis for being ADHD-like, for the ARS activity-impulsivity score, and for the C-BARC energy score. The *p*-values obtained from the Kruskall–Wallis test are provided in Appendix B. Box diagrams of the serotonin and dopamine levels for the dogs with suspected diagnoses of being ADHD-like (vs. the control dogs) and the dogs with scores over and under the median for the different DIAS, Dog ARS, and C-BARQ traits are shown in Figure 1.

The multivariate Lasso regression indicated that the dogs with lower serotonin and dopamine serum concentrations were more likely to be diagnosed as being ADHD-like. The results also suggested that gender, neuter status, and age were significant contributors to the prevalence of ADHD-like symptoms (with higher odds in male, entire, and younger dogs) (Table 4).

Regarding the total DIAS score, dogs with lower serotonin and dopamine serum concentrations were more likely to have scores over the median, aside from males and entire animals. In relation to behavioral regulation, it varied with serotonin concentrations in the same direction as the total DIAS score; the model also indicated that the behavioral regulation scores increased with decreased age. Finally, serotonin and dopamine levels were also linked to aggression, aside from gender, dog activity pattern, and number of dogs previously owned (with male dogs being more likely to have DIAS aggression scores over the median, whereas dogs that performed some activity or dogs whose owners who had previously had dogs were more likely to have scores under the median) (Table 4). No relationship was found between serotonin or dopamine concentration and responsiveness.

Data from the Dog ARS activity-impulsivity indicated that dogs with lower serotonin serum concentrations were more likely to have scores over the median (Table 4). No relationship was found between serotonin or dopamine concentration and Dog ARS inattention.

No relationships were found between serotonin and dopamine serum concentrations and trainability, stranger-directed aggression, owner-directed aggression, dog-directed aggression, stranger-directed fear, separation-related problems, and excitability, as measured by the C-BARQ.

However, the probability of a score being over the median for dog-directed fear increased with decreased serotonin concentrations. It also increased in males, entire animals, animals acquired at less than 2 months old, and those whose owners had no dogs previously. With regard to rivalry, the regression model indicated that dogs with lower serotonin had higher C-BARQ scores; higher scores were also found for females and entire animals. Lower serotonin levels likewise increased the scores for non-social fear and touch sensitivity. Touch sensitivity was also higher in dogs acquired at less than 2 months of age, and it increased as weight decreased. Serotonin concentration was also related to attachment (higher scores at lower concentrations). Further, attachment scores were higher in males and with increasing weight and lower in active dogs. Finally, energy scores were higher in animals with lower dopamine levels and in younger dogs (Table 5).

## 4. Discussion

In humans, ADHD may be associated with dysregulation of the catecholaminergic and serotonergic systems [30]. Moreover, several studies have shown a decrease in serum serotine in ADHD patients [1,29,30,43]. As in human studies, we found that serotonin and dopamine levels were significantly lower in suspected ADHD-like dogs.

One limitation of the study was the possible ADHD-like diagnosis subjectivity. Because of the lack of a gold standard test, it was based on the behavior signs of being ADHD-like. Nevertheless, three validated questionaries were also used. C-BARQ was used to help in the diagnosis of different behavior problems. Furthermore, the relationships between dopamine, serotonin, and the C-BARQ were not based on the clinical subjective diagnosis. In the same way, the Dog ARS is a validated scale that was developed on the basis of a validated human ADHD questionnaire [8], and the DIAS is a validated owner-based questionnaire used to assess impulsivity [32,41,42]. The use of these different scales allowed us to identify a more objective relationship between the different traits and neurotransmitters.

Regarding the serotonergic system, it was previously thought that dopamine is the key regulator of hyperactivity, but research findings from rodent and hamster models have suggested that serotonin is also involved in hyperactivity [24,44,45,46]. Furthermore, several human and animal studies have indicated a link between poor impulse control and a low level of the major serotonin metabolite 5-hydroxyindolacetic acid in cerebrospinal fluid [30]. ADHD patients with oppositional defiant disorder (the most frequent coexisting condition in ADHD, characterized by a repeatable pattern of negativistic and defiant behavior) showed lower serum serotonin levels than pure ADHD patients. Furthermore, some tricyclic anti-depressants, as well as monoamine oxidase inhibitors, which are known to increase serotonergic activity, are apparently efficacious in reducing hyperactivity, impulsiveness, and concentration impairment in ADHD children [30,47].

There are few studies related to impulsivity and serotonin in dogs. The DIAS was previously used to assess urine serotonin and dopamine in impulsive and control dogs. Higher DIAS impulsivity scores were found to be significantly correlated with impulsive behavior (reduced tolerance to delay of reinforcement) and lower levels of urinary serotonin and serotonin/dopamine ratios [32]. Further, Peremans et al. [48] supported the role of the serotonergic system in canine impulsive aggression. Similar results were found in the present study in which dogs with higher DIAS scores showed lower serotonin concentrations. Moreover, the other impulsivity-hyperactivity scale used in this study, the Dog ARS, also related impulsivity-hyperactivity scores with lower serotonin levels. In this line, the clinical effectiveness of fluoxetine, a selective serotonin re-uptake inhibitor, in the treatment of ADHD-like dogs [25,26,27] could be linked with the serotonin effect on impulsivity, hyperactivity, and with the intimate interplay between serotonin and dopaminergic neurotransmission [24].

ADHD is frequently associated with comorbid aggression [14], anxiety, and phobias [49,50,51]. Furthermore, ADHD- like dogs have had strong comorbidities with compulsive behaviors, aggressiveness, and fearfulness [52]. A strong relationship between aggression and low serotonin levels has been demonstrated in different species, including dogs [33,34,36]. Similar results were found in the present study where different aggression scales were related to serum serotonin. The Lasso regression indicated that the dogs with lower serotonin levels were more likely to have scores over the median for the DIAS subscale of aggression and for the C-BARQ rivalry behavioral trait. In relation to the above, the aggressive dogs showed low serotonin levels; however, aggression is one of the symptoms and comorbidities of being ADHD-like precisely because of the tendency to behave impulsively, and it is also related to low serotonin levels. Serotonin likely mediates the tendency to behave aggressively and impulsively in ADHD-like dogs.

Animal models consistently demonstrate that brain serotonin systems are critically involved in the response to stressors, as well as in fear and anxiety generation [53,54,55]. Accordingly, lower serotonin concentrations have been found in dogs showing defensive forms of aggression than in other aggression forms [33], and serotoninergic drugs are used to treat fear and phobias in humans [56,57] and dogs [58,59]. Interestingly, in the present paper, significantly lower serotonin levels were found in the dogs with higher C-BARQ scores for fear, anxiety, and touch sensitivity, and touch sensitivity is related to fearful or wary responses to potentially painful or uncomfortable procedures. Moreover, previous research has shown the relationship between serotonin and attachment. In this sense, early maternal deprivation is related to decreasing 5-hydroxyindoleacetic levels in the cerebrospinal fluid and/or through changes in serotonin transporter gene DNA methylation [60,61,62]. Nevertheless, the association between blood serotonin and attachment in dogs was not previous described. Although the importance of serotonin in the social behaviors of different species has already been studied, is still needed to clarify the relationship between blood serotonin and attachment in dogs.

Since approximately 70% of the children and adolescents with ADHD successfully respond to methylphenidate [1,15] and some ADHD-like dogs respond to this treatment as well [16,17], it appears that dopamine has an important role in ADHD. Some genes of the dopaminergic neurotransmitter system were associated with activity, impulsivity, and attention deficits in dogs [19,20,63,64]. Nevertheless, this is the first study that showed a relationship between ADHD-like dogs and serum dopamine. More specifically, it appeared that the dogs with high activity scores were more likely to show lower dopamine levels. Dopamine functions are related to voluntary movement. Therefore, studies on hamsters indicated that Rovorowsky hamsters (which are more hyperactive than Djungarian hamsters) had lower levels of brain dopamine [45].

No relationship was found between serotonin or dopamine concentration and attention deficits. Furthermore, police German Shepherd dogs (possessing polymorphism in exon 3 of the dopamine D4 receptor gene) showed significantly higher scores in the activity-impulsivity component of the dog-ADHD rating scale than did dogs without this allele. This relationship was not found for the attention deficit component [20]. More studies appear to be necessary in order to deepen the relationship between attention deficits and dopamine and serotonin levels, or perhaps the serum levels of dopamine and serotonin may not be good markers for studying attention deficits.

A higher incidence of human ADHD is described in boys (compared to girls) [1,2]. Comparable results were found in our study where males suffered from being ADHD-like significantly more frequently than females, aside from having higher DIAS scores. Previous research has found similar risk factors for dogs [37,65]. In addition, neutering appeared to be an important risk factor for being ADHD-like, and entire animals were more prone to suffering ADHD-like behaviors than castrated dogs and had higher DIAS scores, as previously described [37,65]. Nevertheless, Zink et al. [66] found that neutering could be a risk factor for hyperactivity, whereas Fadel et al. [67] indicated that castrated animals showed higher DIAS scores. Nevertheless, some owners could try to find a solution for dealing with the condition of their dog by neutering. Hence, the role of sex hormones in ADHD-like behaviors should be studied in more detail.

Interestingly, age was also a significant contributor to the prevalence of ADHD-like symptoms. Furthermore, the DIAS behavior regulation score increased with decreasing age. Other studies have found that hyperactive, impulsive, and inattentive behaviors were much more prevalent in young dogs [9,19,41,65]. Likewise, in humans, ADHD is age-dependent, being more prevalent in youths [4], in spite of the possibility that the condition can remain present in adult humans. Young dogs may appear more hyperactive to inexperienced owners; however, the possible effect of age was used as a control variable in the multivariate analysis.

Dogs that performed some activity and dogs from owners who had previously had dogs were more likely to have scores below the median for DIAS aggression. Other studies observed associations between an owner’s dog experience and a dog’s aggressive behavior [52,54,55,56,57,68,69,70]. It is possible that experienced owners are more aware of the importance of socialization. Previous experience can also help owners to identify a problem at an early stage, when the problem can be treated more efficiently. Furthermore, if the owners had problems with their first dogs, they may be more careful when choosing a new dog [52]. On the other hand, they could have a comparison point if the previous dog was behaving normally. Hence, they might be more aware that something is not normal and refer to it earlier. In addition, they might have more experience in dog training, which could be a part of the improvement in their dog.

Activity patterns had a relationship with the presence of aggression. The complete underlying mechanism behind the effect of exercise in aggression is not clear, but it is known that exercise increases serotonin production both in animals and humans, thus functioning as an antidepressant [71], and it was also related to reduced anxiety [72] and aggression in dogs [70]. In addition, in ADHD-like cases, owners might have difficulties in walking or exercising their dogs. Consequently, the quantity of exercise might be a consequence of the dog being challenging for months, rather than a correlation or a cause.

Finally, the paper highlights the importance of pairing assessments by veterinary specialists using validated tests with avoiding assessments by other professionals. Further, it would be interesting to assess the role of genetics and neurotransmitters in ADHD-like dogs in future studies.

## 5. Conclusions

Our data suggest a strong relationship between serotonin and dopamine and ADHD-like dogs. Furthermore, lower serotonin and dopamine serum concentrations were related to being ADHD-like. Moreover, serotonin and dopamine levels were also linked to aggression, activity, and impulsivity. Decreased serotonin concentrations were also related to fear, attachment problems, and touch sensitivity.

No relationship was found between serotonin and dopamine serum concentrations and attention deficits.

## Figures and Tables

**Figure 1 animals-13-01037-f001:**
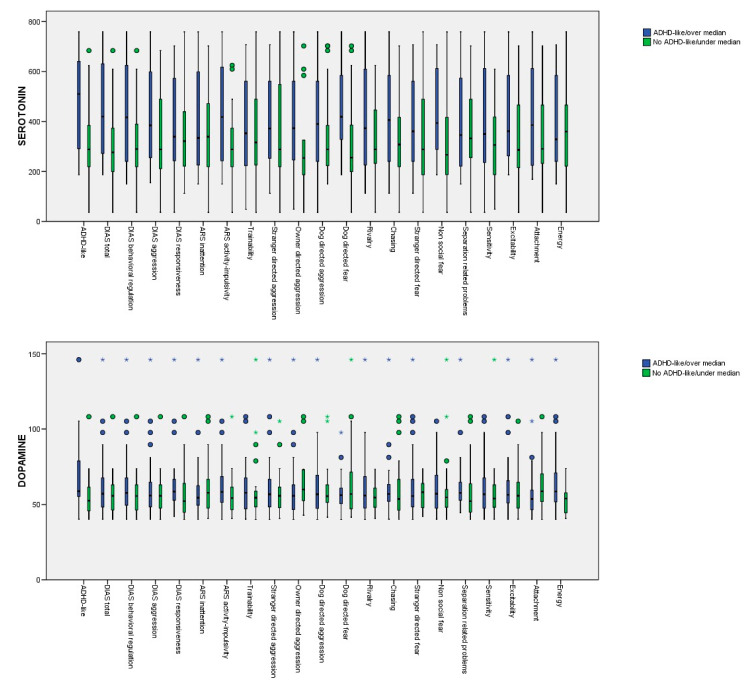
Box diagrams of the serotonin and dopamine levels for the dogs with suspected diagnoses of being ADHD-like (vs. the control dogs) and the dogs with scores over and under the median for the different DIAS, Dog ARS, and C-BARQ traits.

**Table 1 animals-13-01037-t001:** ADHD-like symptoms.

Attention Deficit Symptoms
Other things attract attention easily
Loses interest easily
Difficulty concentrating
Does not pay attention to someone speaking to him/her
Difficulty performing practiced tasks
Easily distracted
**Hyperactivity-impulsivity**
Difficulty maintaining stay
Barks endlessly
Fidgets or in constant motion
Excessive active play/running around
Reacts hastily/anticipates
Cannot wait
Noninhibited bite
Lack of safety
Lack of sleep during the day and frequent waking up and movement during the night
**Aggressive displays**

Modified version of the lists created by Bleuer-Elsner et al., Hoppe et al., Lit et al., and Vas et al. [8,9,25,37].

**Table 2 animals-13-01037-t002:** Descriptive analysis of the studied dog population according to ADHD-like clinical diagnosis.

Variable	Descriptive Features		
	Total (n = 58)	ADHD-Like (n = 36)	Not ADHD-Like (n = 22)
Gender	Data	Data	
Female	29 (50%)	16 (44.4%)	13 (59.1%)
Male	29 (50%)	20 (55.6%)	9 (40.9%)
Neutered			
Yes	42 (72.4%)	28 (77.8%)	14 (36.4%)
No	16 (17.6%)	8 (22.2%)	8 (63.6%)
Age (mean, median, (S.D.), and range)	48.2, 45, (33.9), and 7–156	40.2, 34, (31.1), and 11–121	61.2, 61, (35.0), and 7–156
Age at acquisition (mean, median, and (S.D.))	6.1, 2, and (11.1)	6.2, 2, and (13.3)	5.9, 2, and (6.3)
Weight (mean, median, and (S.D.))	21.4, 22, and (9.4)	21.3, 21, and (8.4)	21.7, 23, and (11.1)
Dog activity patterns			
No activity	54 (93.1%)	34 (94.4%)	20 (90.9%)
Agility/biking/marching/mushing/other	4 (6.9%)	2 (5.6%)	2 (9.1%)
Dogs previously owned			
None	19 (32.8%)	12 (33.3%)	7 (31.8%)
One or more	39 (67.2%)	24 (66.7%)	15 (68.2%)
Habitat			
Rural	19 (32.8%)	11 (30.6%)	9 (40.9%)
Urban	39 (67.2%)	25 (69.4%)	13 (59.1%)

**Table 3 animals-13-01037-t003:** Mean serotonin and dopamine serum concentrations (ng/mL) of the dogs with suspected diagnoses for being ADHD-like (vs. the not ADHD-like dogs) and the dogs with scores over and under the median for the different DIAS, DOG ARS, and C-BARQ traits (different superscripts indicate significant differences). The mean scores obtained for the dogs with scores over and under the median for the different traits are provided, along with the standard deviations (SDs).

	Serum Serotonin Concentration (95% CI)	Serum Dopamine Concentration (95% CI)	Mean Score (SD)
ADHD-like			
Yes	311.91 (257.36–366.45) ^a^	55.49 (50.51–60.47) ^a^	
No	477.06 (391.20–562.92) ^b^	68.68 (57,54–70,81) ^b^	
DIAS total score			
Under median	449.28 (376.98–521.59) ^a^	63.90 (54.66–73.14)	0.46 (0.07)
Over median	304.29 (240.75–367.83) ^b^	57.41 (51.49–63.39)	0.73 (0.08)
DIAS behavioral regulation score			
Under median	428.56 (353.75–503.37) ^a^	63.99 (55.14–72.85)	0.37 (0.10)
Over median	321.18 (255.63–386.73) ^b^	57.04 (50.87–63.22)	0.75 (0.11)
DIAS responsiveness score			
Under median	389.56 (315.02–464.10)	64.32 (55.46–73.18)	0.32 (0.08)
Over median	364.01 (291.23–436.80)	57.25 (50.57–63.92)	0.65 (0.20)
DIAS aggression score			
Under median	414.08 (342.37–485.78) ^a^	63.16 (53.60–72.71)	0.58 (0.11)
Over median	334.82 (259.17–410.47) ^b^	57.57 (51.66–63.49)	0.81 (0.08)
Dog ARS activity-impulsivity score			
Under median	432.10 (360.58–503.62) ^a^	64.14 (55.52–72.77) ^a^	2.90 (1.70)
Over median	303.04 (240.17–365.91) ^b^	56.57 (50.38–62.77) ^b^	10.1 (3.61)
Dog ARS inattention score			
Under median	402.22 (328.84–475.59)	60.28 (52.01–68.56)	4.16 (3.14)
Over median	347.45 (275.34–419.55)	61.20 (53.76–68.64)	14.77 (4.15)
C-BARQ trainability			
Under median	381.53 (310.88–452.18)	60.48 (54.46–66.51)	2.15 (0.36)
Over median	371.32 (293.90–448.74)	61.04 (50.44–71.64)	3.13 (0.30)
C-BARQ stranger-directed aggression			
Under median	394.32 (331.29–457.34)	62.75 (54.16–71.35	0.04 (0.07)
Over median	356.56 (271.37–441.75)	58.25 (51.71–64.79)	0.98 (0.81)
C-BARQ owner-directed aggression			
Under median	401.08 (344.04–458.13) ^a^	58.98 (53.04–64.91)	0.02 (0.07)
Over median	303.90 (192.38–415.42) ^b^	67.35 (52.94–81.76)	0.62 (0.55)
C-BARQ dog-directed aggression			
Under median	394.89 (327.73–462.05)	61.66 (53.78–69.53)	0.15 (0.03)
Over median	348.81 (267.45–430.17)	59.39 (51.76–67.02)	1.56 (0.91)
C-BARQ dog-directed fear			
Under median	449.32 (378.88–519.75) ^a^	57.36 (51.96–62.75)	0.21 (0.24)
Over median	318.03 (250.60–385.99) ^b^	63.71 (54.44–72.99)	1.38 (0.78)
C-BARQ rivalry			
Under median	421.10 (338.25–503.94)	61.97 (52.81–71.13)	0.02 (0.08)
Over median	331.96 (258.93–404.99)	55.61 (50.91–60.31)	0.94 (0.54)
C-BARQ chasing			
Under median	414.29 (52.96–70.87)	61.92 (52.96–70.87)	0.66 (0.55)
Over median	344.29 (52.64–66.80)	59.72 (52.64–66.80)	2.78 (0.78)
C-BARQ stranger-directed fear			
Under median	397.87 (336.00–459.75)	62.76 (54.40–71.12)	0.05 (0.10)
Over median	344.20 (253.34–435.06)	57.34 (52.59–62.08)	1.58 (1.01)
C-BARQ nonsocial fear			
Under median	438.03 (366.61–509.45) ^a^	61.49 (54.85–68.13)	0.27 (0.19)
Over median	319.77 (251.00–388.54) ^b^	59.91 (50.79–69.03)	1.54 (0.77)
C-BARG separation-related problems			
Under median	387.91 (308.59–467.23)	63.22 (54.84–71.60)	0.16 (0.17)
Over median	366.43 (297.99–434.87)	58.30 (50.94–65.67)	1.18 (0.48)
C-BARQ sensitivity			
Under median	407.42 (340.31–474.52)	60.95 (54.84–67.05)	0.23 (0.21)
Over median	321.65 (246.73–396.58)	60.26 (48.60–71.92)	1.34 (0.70)
C-BARQ excitability			
Under median	414.16 (346.02–482.31)	62.92 (54.08–71.76)	1.09 (0.36)
Over median	336.64 (260.02–413.27)	58.24 (51.84–64.64)	2.39 (0.50)
C-BARQ attachment			
Under median	419.28 (344.33–494.23)	59.31 (50.48–68.14)	1.36 (0.45)
Over median	334.29 (265.56–403.02)	62.17 (55.41–68.94)	3.08 (0.62)
C-BARQ energy			
Under median	392.08 (326.48–457.69)	65.62 (57.16–74.08) ^a^	2.08 (1.12)
Over median	354.84 (270.15–439.54)	53.81 (49.13–58.49) ^b^	3.81 (0.72)

**Table 4 animals-13-01037-t004:** Results of Lasso regression models for the effects of serotonin and dopamine serum concentrations in dogs on ADHD-like diagnosis (yes/no) and scores (over and under the median) for the different DIAS and DOG ARS traits.

Coeffcients	ADHD-Like R^2^ = 0.4698	DIAS Total R^2^ = 0.2119	DIAS BR R^2^ = 0.2134	DIAS A R^2^ = 0.1716	Dog ARS AI R^2^ = 0.08921
Serotonine	−0.0010	−0.0007	−0.0005	−0.0005	−0.0004
Dopamine	−0.0057	−0.0013			
Gender Male *	0.0609	0.0411	0.1427	0.0207	-
Neutered Entire **	0.0630	0.0230	0.0317	-	-
Age	−0.0025	-	−0.0002	-	-
Age at acquisition				-	-
Weight	-	-	-	-	-
Dog activity pattern ***	-	-	-	−0.3389	-
First dog	-	-	-	−0.0180	-

* Female is the base; ** neutered is the base; *** no activity is the base.

**Table 5 animals-13-01037-t005:** Results of Lasso regression models for the effects of serotonin and dopamine serum concentrations in dogs on scores (over and under the median) for the different C-BARQ traits.

Coeffcients	Dog-Directed Fear R^2^ = 0.4380	Rivalry R^2^ = 0.25117	Nonsocial Fear R^2^ = 0.1399	Sensitivity R^2^ = 0.2143	Attachment R^2^ = 0.2284	Energy R^2^ = 0.2638
Serotonine	−0.0006	−0.0006	−0.0004	−0.0001	−0.0005	
Dopamine						−0.0044
Gender Male *	0.3686	−0.0129			0.1800	
Neutered Entire **	0.1095	0.1564				
Age						−0.0046
Age at acquisition	−0.0908			−0.1593		
Weight				−0.0107	0.0028	
Dog activity pattern ***					−0.0406	
First dog	0.2281	−0.2629			−0.2646	

* Female is the base; ** neutered is the base; *** no activity is the base.

## Data Availability

The data presented in this study are available on request from the first author/corresponding authors.

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
