# Peer review of "Serotonin and Dopamine Blood Levels in ADHD-Like Dogs"

_animals, 2023, doi:10.3390/ani13061037_

Round 1

Reviewer 1 Report

PRINCIPAL STRENGTHS OF THE MANUSCRIPT:

The paper presents findings that may help to better understand the neurophysiology of the ADHD-like disorder in dogs. The rationale and aims of the study are clear, and the authors provide a sound human literature review that serves as the basis of the current study.  I think that this paper could serve as a basis for further research on the neuropathology of this behavioural problem. I think that a strong point of the paper is that the authors checked the neurotransmitters levels.  I enjoyed reading the paper and I appreciate its short length of it.  However before I can recommend it for publication, different issues deserve clarification, especially regarding the statistical analyses. For this reason, I am suggesting a major revision. Good luck to the authors and looking forward to seeing the revised version of the manuscript.

Mayor comments

I am confident that all my issues with the manuscript are due to a lack of written information on the author’s part and not due to a wrong methodology. I really recommend they include everything that is needed to understand the procedure.

Methods

Ethical statements and approval are missing from the methods.

Subjects.

It is unclear to me which final subjects were included in the analyses and how was this inclusion criterion done. Were the questionnaires performed first, and based on the scores were the animals invited to the behavioural consultation? Alternatively, on the other hand, was behavioural assessment performed first after the tutor brought the animal for the consultation?

Furthermore, Have every dog included in the questionnaire performed both parts of the procedure (behavioural assessment and questionnaires)?

How many subjects were assessed in every center? This information is lacking and it would be very important, as differences in the procedure (including place) could have an impact on the given results (eg. if those dogs sampled in one specific location have very similar serum results this could be somehow related with the procedure rather than to the subject itself)

The final subject number is 36 (No 58) and those 36 are supposed to be ADHD-like dogs. I would like to see the demographics of the 36 dogs  (age, gender, etc.). This data is not clear as the authors provide the data from the whole sample and not of final included subjcts.

Importantly It is not clear which dogs belong to the control group, (noN-ADHD like dogs). Are those the 22 remaining dogs?? Demographics?

To better address some of the above comments, I would recommend the authors provide as supplementary information all the subjects' details (sex, age, BREED)

I think that Table 1 is a very good table, I really like all the parameters that the authors include BUT I would ask then to do it with the final subject sample.

Besides, It will be very useful to have a table with all the subjects and their measurements for each questionnaire to see them all together (eg. dog1, yes/no, dias score, cbarq score etc)

Dog Assessment

Am I understanding correctly that higher scores in the questionnaires mean that the behaviour problem is more severe or that it appears more often?

Statistical analyses

136-the medians of what? Of each individual? How did the authors calculate the cut-offs?

136- From here I understand that the scores given a 1 (binary variables) are those over/above the median is that correct? And 0 those scores below the median. Thus dogs that are more likely to have behaviour problems, or more often are given a 1 in the binary variables for a given score.

The authors should explain why they decided to transform the questionnaire scores into binary variables.

Besides is not clear to me how many subjects are included in each group. The authors only provide the number of the Yes/No based on behav. assessment,  how many dogs are belonging to the Dias over the median, and under? Etc, etc.

Importantly, regarding the binary variables. We need to know:

1-How was the scale of the measurement?? For example, the authors say that the C-barq questions are scaled on a 1-4 scale. Is this the same scale as the overall? How did the authors calculate the overall of all questionnaires? We need to know the overall scale for each measurement.

2- Because then we need to know the exact results of the dogs included in each group. 

For example: if the overall scale goes from 1-10, but from your results, the dogs go from 1-3, and you provide a cut-off in the median, you will have your under and over median results BUT those dogs would ALL of them in the lower scale. We don’t know if this is the case because the authors don’t provide any explanation regarding this.

So I would ask them to provide the overall scale and the mean and SD of dogs that are under the median and over, to see their real ratings.

Results

155-158. The authors should explain why they chose Kruskal-Wallis est. I understand that the data was not fitting into a normal distribution, did the authors try any transformations? Besides, the authors should provide the results of the Kruskal-Wallis test (in a supplementary information file would be ok)

Table 2

I would recommend the authors include figures in the manuscript with the individual data points of all subjects as without them is very difficult to understand the variability among each group and the validity of the given results.

Furthermore, If I understood correctly the transformation of the scores on binomial variables, those animals with lower scores were given a 0, and those with higher (and a suspect of ADHD-like symptoms) were given a 1, those animals OVER the median are the more likely to have ADHD-like and those UNDER are the normal dogs. Then the findings reported in Table 2 seem the opposite of what the authors are claiming. Subjects with higher scores on the questionnaires have higher levels of both neurotransmitters. I believe that this is the result of a misleading explanation from the authors and not the actual fact that normal dogs have lower levels of serotonin and dopamine (which would mean that the results show the opposite of the authors' hypothesis). I would recommend explaining carefully how the authors transformed the scores to binomial variables, what means, and an overall careful check up of the results.

The authors should explain why did they use a Lasso regression and not other statistical models (like BiGLM).

Table 3

Report the whole results of the model and not only the confidence interval. P values and SD are needed,

I also have another major question regarding the results. Why didn’t the authors perform correlations between the different scores? I think It will be much more informative to know if all those scores are measuring the same or not. (between the behav assessment, the Dias, the C-barq etc)

I encourage the authors to provide these results. If there are no correlations or weak correlations the authors should discuss this carefully.

 Discussion

The discussion lacks a limitation paragraph and a further directions paragraph.

Besides I would recommend the authors to  re-check their dsicussion after correcting the statistics and resutls part, and change it accordingly.

Minor comments

94-Typo on behavioural. Add behavioural examination.

134- R packages are missing 

227- see my above comment. Where is this result written? Based on table 2 higher DIAS scores showed HIGHER serotonin concentrations.

Author Response

Dear Review, 

Thank you for yours comments.

Please, get the document attached with a point-by-point response.

Best regards.

The authors

Reviewer 2 Report

This original research is of interest for the field, because having biological markers for behavioral conditions is lacking.

However, the link between serotonin and ADHD like condition is not clearly explained from a neurophysiological point of view. ADHD has been shown to depend on dopaminergic and adrenergic dysregulation in specific areas of the brain (Stahl, 2013). And ISRS effects can be better explained too as why they improve humans or dogs with such conditions. Also, the authors state that ADHD seems to solve itself in adults. But a major reference in human medicine (Hoogman, 2017) shows that it is untrue and still present in most adults too.
The link between the serum dosages realized and ADHD condition is also missing from a perspective of possible practical implications for clinicians.

Moreover, in the study design, even if most of the experimental design is well explained, the way specialists establish their symptomatic ADHD-like behaviors as well as the dog inclusion criteria in remain unclear. The authors state a very high prevalence of ADHD-like dogs and this should be explainable : were the authors trying to have mostly ADHD-like dogs ? or balanced groups between ADHD-like and non ADHD-like dogs?

Also, concerning the dog inclusion criteria, it should be discussed whether fear and anxiety are symptoms of ADHD or other conditions. If they are of other conditions, then the low serotonin level might not be a marker of ADHD, but of other conditions or more likely of many different conditions. Hence, measuring low serotonin level will not help neither to diagnose ADHD, nor to treat it.

The term of “ADHD behavior” seems sometimes inappropriate since the literature rather shows that it is a neurodevelopmental condition which is recognized as a disease in humans. This should be made clearer.

SPECIFIC COMMENTS

Line 42 : the paper from hoogman and colleagues (2017), a major meta analysis in human psychiatry, should be included because it shows that it does not concern only childhood. The difference with normal subjects remains all life long and can be seen in MRI, even though it is indeed most important during childhood.

Lines 51-52 : the neurophysiology of ADHD should be better explained. Again, Hoogman and colleagues’ paper explains which part of the brain exhibits abnormal functioning. This disease concerns specific area in the brain and just talking about neurotransmitters in general might not be enough to understand how it works. Explaining these specific mechanisms thus will render the paper stronger.

Lines 53-59 : same comments for the dopamine and the serotonin part. Several references are provided, but it is hard to understand dopamine role just by reading this paragraph. Dopamine is involved in motivation but it could be explained how a high motivation can lead to ADHD symptoms.

Lines 67-73 : yes low levels of serotonin are found in many disorders and this very general correlation does not explain the exact mechanism. As stated by the authors, serotonin is a very general inhibitor which might explain why it is found correlated to many psychiatric conditions but might not be specific enough to diagnose all these diseases. Any condition could respond to serotonin treatments for the same reason : it will give more inhibition capacities to the individual, whatever the exact condition is. As a matter of fact SSRI are used in most psychiatric diseases with at least partially good results. The role of serotonin in modulating most behaviors and, therefore, its lack of specificity should be better explained.

Lines 75-77 : The aim is stated but the link between these serum levels and ADHD needs to be established better (see upper comments)

Lines 93-98 : It is very unclear what are the symptoms used to establish the clinical diagnosis. A reference to one of the clinical textbooks that can be used for this would improve the manuscript. Is it Pageat symptoms, Overall or Landsberg ? Putting tThe exact list that permitpermitted the clinician to establish the diagnosis is needed.

62% of the dogs were diagnosed with ADHD which is way over general prevalence in dogs. Were they recruited with specific symptoms already ? how the authors explain such high number?. Did they look for working dogs, where the disease might have been selected to have dogs that are very impulsive and unable to stop when engaged in  a behavior (a paper from Schilder and colleagues from 2004  : “Training dogs with help of the shock collar: short and long term behavioural effects » could be added because they underline this unethical selection in working dogs)

Concerning comorbidities with phobias, it would have been good to exclude possible comorbidities for this study since we are looking for specific markers of ADHD.

Lines 156-157 : the tests never show the results of P value hence it is really difficult for the reader to see how significant the results are. I would suggest to use different tables or graphics to make it more relevant. Also, the P values should be written somewhere.

The tables seem correct but are uneasy to read and understand. A phew graphics would have been helpful and there is not a single P value in the article which is a huge lack to see how significant the results are.

I don’t know if the authors tested the correlation between severity of the score and low levels of serotonin but this would be interesting to know because then it could be considered as a marker of severity.

Lines 209-216 : yes, serotonin inhibits most behaviors, so as stated before, it should be seen as a way more larger regulatory, which explains why we can find low levels of serotonin in many diseases. The authors should be cautious here because it shows that serotonin level in serum cannot be a good biomarker for these diseases.

234-242 : again, these are all symptoms that will be more seen if the subject lacks general control and inhibition capacities, which is exactly the role of serotonin. This should be used to explain why this low level is associated with all of these symptoms. But the research in this paper focuses on ADHD which is a specific condition. The discussion should include the fact that not taking pure profiles might have compromise the results. (if you want to study a specific carcinoma and take multiple diseases in your sample this will give unclear results).

255-257 : yes and it should be discussed. How can dopamine level be linked to overall more activity.

262-263 : yes or maybe the serum level cannot be a good marker to study this further. This should be stated in the discussion. These behavioral conditions discussed have all been linked to specific dysfunctions in the brain and maybe the serum levels will never be a good marker.

Lines 270-272 : The discussion concerning neutering lacks a part of reasoning. It is known that any behavioral condition leads owners to try neutering as a possible solution for behavior. So the link between sexual hormones and having ADHD might be due to the fact that owners (who decide to neuter the dog or no) try to find a solution to deal with the condition of their dog.

Line 277 : again, it is also shown that the condition remains present in adult humans.

Lines 284-285 : owners that already had a dog have a comparison point if the previous dog was behaving normal. Hence, they might be more aware that something is not normal and consult earlier. Also, they might be more experimented to train their dog, which could be a part of the improvement of the dog. ADHD treatment consists in behavioral therapy in addition to medical treatment and many trainers can provide good exercises to improve the dog impulsivity and general lack of control. This should be discussed as a possible explanation.

Lines 287-289 : the discussion could include the fact that owners might have difficulties with their challenging ADHD-like dog. Consequently, the quantity of exercise might be a consequence of the dog being challenging for months, rather than a correlation or causality. Authors should be more complete on the possible explanations given to interpret their results.

In the conclusion the authors should state that the results are not very specific because so many conditions might be linked to low levels of serotonin and dopamine. These serum levels might not be a suitable marker for ADHD but a larger one involved in most behavioral conditions.

Author Response

(The authors gave the same response as above.)

Reviewer 3 Report

My comment in the attached file

Author Response

(The authors gave the same response as above.)

Round 2

Reviewer 1 Report

I have read now the revised version of the manuscript and first of all, I thank the authors for answering my questions and implementing some changes. However, there are still several issues that concern me and should be solved before I can recommend this article for publication.

However, I still believe this article could constitute a very important piece in the ADHD research field, so I would really encourage the authors to do some fine-tuning of the manuscript.

Good luck wiht the new revision

Major concerns

1-I am really thankful for the clarification regarding the use of the Lasso regression and the Kruskal_Wallis that the authors provided in their revised version. However, now that the authors provide the Kruskal_wallis results in my concern is that I can not find if the authors performed a correction for multiple correlations. Given the amount of correlation with a single measurement (the dopamine and the serotonin levels) this seems a really needed statistical test to perform.

2-Furthemore my other concern lies in the actual discussion (which i have already mentioned in the previous revision) as it is not very straight forward and there are several critical points that the authors should discuss regarding their results. For instance, I would like to see a discussion about the different results that the author got from the Kruskal-Wallis compared with the Lasso regression, for example, based on the Lasso dopamine has an effect on the total Dias score however based on the nonparametric test the 2 groups the authors got were not different. These kinds of results should be discussed in the paper giving the reader suggestions and possible explanations. Besides this, some results were reported but not discussed at all (eg. attachment results from the Lasso regression). Furthermore, several paragraphs of the discussion are more suitable for an intro and this makes the discussion rambling and unfocused. I think that the authors have a very nice collection of results and expending a bit more space in carefully discussing them would increase the quality of the manuscript

Minor comments

First of all, I highly recommend the authors perfom a careful proofreading of their manuscript. The text in its current form has several typos, English misuses, and editing mistakes that increase the difficulty of reading it.

52- because of

49-54- The information given in this paragraph stands by only one reference. I recommend the authors add more or slightly tone down the affirmation in only one reference is available.

75-genetic markers in humans lacks a reference.

80- Place this sentecence in the above paragraph or delete it.

91-94. What did the authors want to say with„ Selective serotonin reuptake inhibitors are clinically effective” Effective for treating ADHD I assume based on the next sentence on dogs, but I would suggest the authors rewrite this part to make it more clear.

151-152- I thank the authors to add the scale of the DIAS scoring system. However, this is not very informative as at the end the scores are transformed on a scale from 0-1. For ease for the reader, I suggest the authors provide this information.

196-there are several independent variables

214-box plot

Figure 1- I appreciate that the authors include a plot in the current manuscript. However, there is information lacking to understand the figures. Why are there significant stars in the dopamine plot and not in the serotonin? What do the stars mean? I mean if the intention was to give us the result of the comparison of both groups the stars are misplaced. Please, provide an explicative legend.

280-290- this paragraph is rather an introduction paragraph. I would suggest the author move any information that they believe is important for the research to the introduction and in the discussion focus on discussing their results.

294-297- Similar remark as in the previous paragraph, and this fact was already explained in the introduction.

299- 302- This is the actual result so I would ask the author to discuss it.

363- I would be very cautious about any strong conclusion on this result as the dogs that perfom some activity are only 4 vs 54 (based on your table) so this result could be even related to individual variability.

387-were

Supplementary data sheet- Thanks for providing this as having the data available is always a sign of transparency. However the data_sheet in its current form is difficult to understand, It would be great if the authors can provide a legend to understand which are the variables that they are reporting ( at least a legend of the most important ones as I understand that because this was mostly a questionnaire study the authors have many variables) Furthermore, I encourage the authors to translate to English their data set.

Author Response

Major concerns

1-I am really thankful for the clarification regarding the use of the Lasso regression and the Kruskal_Wallis that the authors provided in their revised version. However, now that the authors provide the Kruskal_wallis results in my concern is that I can not find if the authors performed a correction for multiple correlations. Given the amount of correlation with a single measurement (the dopamine and the serotonin levels) this seems a really needed statistical test to perform.

One of the advantadges of Lasso regression is precisely that it performs well under collinearity the used parameters:

Srisa-An, C. (2021, August). Guideline of collinearity-avoidable regression models on time-series analysis. In 2021 2nd International Conference on Big Data Analytics and Practices (IBDAP) (pp. 28-32). IEEE.

Abdella, G. M., & Shaaban, K. (2021). Modeling the impact of weather conditions on pedestrian injury counts using LASSO-based poisson model. Arabian Journal for Science and Engineering46, 4719-4730.d

2-Furthemore my other concern lies in the actual discussion (which i have already mentioned in the previous revision) as it is not very straight forward and there are several critical points that the authors should discuss regarding their results. For instance, I would like to see a discussion about the different results that the author got from the Kruskal-Wallis compared with the Lasso regression, for example, based on the Lasso dopamine has an effect on the total Dias score however based on the nonparametric test the 2 groups the authors got were not different. These kinds of results should be discussed in the paper giving the reader suggestions and possible explanations. Besides this, some results were reported but not discussed at all (eg. attachment results from the Lasso regression). Furthermore, several paragraphs of the discussion are more suitable for an intro and this makes the discussion rambling and unfocused. I think that the authors have a very nice collection of results and expending a bit more space in carefully discussing them would increase the quality of the manuscript

A certain discrepancy between the multivariate and univariate analysis is logical and to be expected since the control variables are sometimes not distributed homogeneously between the groups to be compared (for this reason, perhaps it is not appropriate to lengthen the article to discuss two different ways of generating results that are expected to have some discrepancies in any type of approach). That is why all the control variables that could be accurately collected were included and, in this way, obtain the most accurate coefficients possible in the Lasso regression.

Nevertheless the sentence, “The univariate approach indicated significant differences in serotonin serum concentrations for C-BARQ owner direct-aggression” was removed from the discussion in order to focus it tine the multivariate analysis.

The article is very long, so we prefer to focus the discussion on the results related to ADHD-like.

Minor comments

First of all, I highly recommend the authors perfom a careful proofreading of their manuscript. The text in its current form has several typos, English misuses, and editing mistakes that increase the difficulty of reading it. 

52- because of

„Of” was added to the text

49-54- The information given in this paragraph stands by only one reference. I recommend the authors add more or slightly tone down the affirmation in only one reference is available.

Some references were added to the text

75-genetic markers in humans lacks a reference.

Some references were added to the text

80- Place this sentecence in the above paragraph or delete it. 

This sentence was move to the above paragraph

91-94. What did the authors want to say with„ Selective serotonin reuptake inhibitors are clinically effective” Effective for treating ADHD I assume based on the next sentence on dogs, but I would suggest the authors rewrite this part to make it more clear. 

„for treating ADHD” was added tot he sentence

151-152- I thank the authors to add the scale of the DIAS scoring system. However, this is not very informative as at the end the scores are transformed on a scale from 0-1. For ease for the reader, I suggest the authors provide this information.

Although the DIAS results were transformed into a binary variable, the authors consider it necessary to clarify the methodology of the questionnaire. Likewise, then it is clarified in material and methods how to proceed to the transformation into a binary variable

196-there are several independent variables

Are was added to the sentence

214-box plot

Figure 1- I appreciate that the authors include a plot in the current manuscript. However, there is information lacking to understand the figures. Why are there significant stars in the dopamine plot and not in the serotonin? What do the stars mean? I mean if the intention was to give us the result of the comparison of both groups the stars are misplaced. Please, provide an explicative legend.

The asterisks are extreme values in the distribution, there are none in sertonin, that is why they do not appear.

We found two individuals with high dopamine values that appear repeated with an asterisk in the box plot.

280-290- this paragraph is rather an introduction paragraph. I would suggest the author move any information that they believe is important for the research to the introduction and in the discussion focus on discussing their results.

Other reviewer asked to add these sentences to the discussion.

294-297- Similar remark as in the previous paragraph, and this fact was already explained in the introduction.

Other reviewer asked to add this sentence to the prargraph.

299- 302- This is the actual result so I would ask the author to discuss it.

This paragraph was changed in order to clarify better the discussion

363- I would be very cautious about any strong conclusion on this result as the dogs that perfom some activity are only 4 vs 54 (based on your table) so this result could be even related to individual variability.

Despite the few individuals, the difference found was significant, which is why we named it. It is more difficult to find significant differences with small groups, but if they do appear, it is ruled out that they are due to chance.

387-were

" Was" was changed with were

Reviewer 2 Report

The authors did a huge job of reviewing on this paper and it has been improved a lot.

I believe the manuscript has been  sufficiently improved to warrant publication in Animals.

Author Response

Thank you for your comments.

Reviewer 3 Report

This amended version is more complete, please note that in line 115 there is a typo, ADHAD instead of ADHD. since you mention in lines 130-134 that dogs with phobias, hyperactivity associated to lack of exercise, impulsivity as result of previous learning and organic pathologies were excluded, how many dogs were originally examined? This would be interesting to know. 

I think that mentioning the importance of pairing assessment by specialists and the results of validated tests vs assessment by other professionals (non specialised vets, nurses and trainers) and the role of genetics should be added as suggestion for future studies.

Author Response

This amended version is more complete, please note that in line 115 there is a typo, ADHAD instead of ADHD. since you mention in lines 130-134 that dogs with phobias, hyperactivity associated to lack of exercise, impulsivity as result of previous learning and organic pathologies were excluded, how many dogs were originally examined? This would be interesting to know. 

ADHAD was changed with ADHD.

This sentence was changed: “5 dogs with suspected attention deficit related to phobias, hyperactivity that could be associated with lack of exercise and impulsivity that could be originated from previous learning, as well as patients with organic pathologies (including chronic pain) were not included in the final sample of 58 animals”

I think that mentioning the importance of pairing assessment by specialists and the results of validated tests vs assessment by other professionals (non specialised vets, nurses and trainers) and the role of genetics should be added as suggestion for future studies.

This paragraph was added: Finally, the paper highlights the importance of pairing assessment by veterinary specialists using validated tests while avoiding assessment by other professionals. Besides, it would be interesting to assess the role of genetics and neurotransmitters in ADHD-like in future studies

Round 3

Reviewer 1 Report

I am thankful for the authors’ answers and implementations in the manuscript.

However, I still have some concerns with the answers.

I agree with the authors that their dog activity result pattern should be discussed, as it is related to their hypothesis. However, their explanation is not completely accurate: finding a significant result with four individuals can be due to a random effect. I suggest the authors add a sentence similar to this to their discussion paragraph: sample size is relatively small, warranting caution in generalizing the results.

On the other hand, reporting results and not discussing them is not appropriate. I agree with their stamen about the length of the article and that they want to focus on ADHD-related results. Then I suggest them:  a) either removed the non-discussed results from the conclusion and abstract (no discussed result shouldn’t be on the conclusion)  or b) discussed them shortly in a supplementary info file.

Minor comments

There are still some typos throughout the article. For example:

199- "Independent variables" is written twice.

I still encourage the authors to provide a supplementary data table completely in English and with a legend to understand the variables. This would assure the clarity and transparency of their provided data set.

Author Response

Dear reviewer, 

Thank you for your comments.

You will find the authors point-by-point response here bellow:

I agree with the authors that their dog activity result pattern should be discussed, as it is related to their hypothesis. However, their explanation is not completely accurate: finding a significant result with four individuals can be due to a random effect. I suggest the authors add a sentence similar to this to their discussion paragraph: sample size is relatively small, warranting caution in generalizing the results.

In this sense, the paper was revised by a certificated statistic. In small sample the differences will have to be larger in order to reach statistical significance. So the, the probability that the results could be due to a random effect are minor to 0,05 %. In fact, the current trend in animal experimentation is to use the smaller samples as possible.

On the other hand, reporting results and not discussing them is not appropriate. I agree with their stamen about the length of the article and that they want to focus on ADHD-related results. Then I suggest them:  a) either removed the non-discussed results from the conclusion and abstract (no discussed result shouldn’t be on the conclusion)  or b) discussed them shortly in a supplementary info file.

These lines were added to the text:

Interestingly, in the present paper, significant lower serotonin levels were found in dogs with higher C-BARQ scores for fear, anxiety and touch sensitivity; the last is related to fearful or wary responses to potentially painful or uncomfortable procedures. Moreover, some previous research showed the relationship between serotonin and attachment. In this sense, early maternal deprivation is related to decreasing 5-hydroxyindoleacetic levels in the cerebrospinal fluid and/or through changes in serotonin transporter gene DNA methylation [62–64]. Nevertheless, the association between blood serotonin and attachment in dogs was not previously described. Although the importance of serotonin in social behavior of different species was already studied, is still needed to clarify the relationship between blood serotonin and attachment in dogs.

199- "Independent variables" is written twice.

“Independent variables” was removed from the text